# Rapid optimisation of fragments and hits to lead compounds from screening of crude reaction mixtures

Lisa M. Baker [1,7], Anthony Aimon [2,3,7], James B. Murray[1], Allan E. Surgenor [1], Natalia Matassova [1], Stephen D. Roughley [1], Patrick M. Collins [2], Tobias Krojer [4], Frank von Delft [2,3,4,5] & Roderick E. Hubbard [1,6 ✉]

Fragment based methods are now widely used to identify starting points in drug discovery and generation of tools for chemical biology. A significant challenge is optimization of these weak binding fragments to hit and lead compounds. We have developed an approach where individual reaction mixtures of analogues of hits can be evaluated without purification of the product. Here, we describe experiments to optimise the processes and then assess such mixtures in the high throughput crystal structure determination facility, XChem. Diffraction data for crystals of the proteins Hsp90 and PDHK2 soaked individually with 83 crude reaction mixtures are analysed manually or with the automated XChem procedures. The results of structural analysis are compared with binding measurements from other biophysical techniques. This approach can transform early hit to lead optimisation and the lessons learnt from this study provide a protocol that can be used by the community.

[1] Vernalis (R&D) Ltd, Granta Park, Cambridge CB21 6GB, UK. [2] Diamond Light Source Ltd., Harwell Science and Innovation Campus, Didcot OX11 0QX, UK. [3] Research Complex at Harwell (RCaH), Rutherford Appleton Laboratory, Didcot OX11 0FA, UK. [4] Structural Genomics Consortium (SGC), University of Oxford, Oxford OX3 7DQ, UK. [5] Department of Biochemistry, University of Johannesburg, Auckland Park 2006, South Africa. [6] YSBL, University of York, Heslington, York YO10 5DD, UK. [7] These authors jointly supervised this work: Lisa M. Baker, Anthony Aimon. ✉email: roderick.hubbard@york.ac.uk

Fragment-based discovery is now established as an effective method in providing starting points to develop compounds that can inhibit or activate the function of a protein[1–3]. The central premise is that small (usually < 18 heavy atoms) compounds have sufficient chemical functionality to bind, but are small enough to fit into site(s) on a protein[4]. This contrasts with the difficulties of identifying compounds that bind (hits) from screening of libraries of larger compounds where additional atoms on a compound may sterically block binding. There are currently four molecules approved for treating patients[5–8] that were discovered with fragment-based methods and many other clinical trials underway[2].

The first step in fragment-based discovery is to screen a library of fragment molecules against the protein target. There is a large literature and experience of using a range of biophysical and functional assays to perform such screening[2,9]. Most practitioners use a biophysical assay (such as surface plasmon resonance (SPR), protein or ligand-observed NMR or thermal shift measurements) to detect binding of a fragment on a target and then subsequently attempt to determine its structure (usually by X-ray crystallography). An alternate approach (first demonstrated by Abbott[10], but successfully promoted by Astex[11]) is to screen directly by determining the X-ray crystal structure of the protein soaked with the fragment(s). This requires the protein to form crystals with an accessible binding site and that can survive the high concentration of the fragment. This approach has the advantage of being able to detect very weak binding, and directly characterises the fragment-binding pose. However, it has until recently been very challenging to prepare the large number of soaked individual crystals, and to handle the amount of data collection and processing required to analyse the many thousands of diffraction datasets needed to screen individual compounds in a fragment library. The recent development of the XChem facility at the Diamond synchrotron[12–14] has greatly improved the throughput, reliability and feasibility of performing such crystallographic screening of fragment libraries, and now screening of 1000-member fragment libraries can be completed (and data analysed) within a few days.

For most proteins, fragment-based screening can identify many 10 s (and in some cases, hundreds) of fragment hits[15–18]. The issue then is identifying which of the hits to optimise, or how best to analyse the structural information given by multiple binding events. The usual practice is to explore the structure–activity relationships (SAR) of the fragments by accessing (from commercial vendors, from internal library collection or by directed synthesis) compounds that are similar to the fragment (so-called nearest-neighbour compounds). These compounds can then establish which compound(s) are suitable for structure-guided fragment to lead optimisation.

Recently, we have developed an additional approach for optimisation of fragments to hits and subsequently to lead compounds[19,20], which has also been exploited by others[21]. For a simple one-step dissociation process, the thermodynamics and kinetics of ligand (L) association with a protein (P) can be represented as in Eqs. (1) and (2)

$$PL \xrightleftharpoons[k_{off}]{k_{on}} P + L \qquad (1)$$

$$K_D = \frac{k_{off}}{k_{on}} = \frac{[P]\ [L]}{[PL]}, \qquad (2)$$

where $K_D$ is the equilibrium dissociation constant (often referred to as affinity) and $k_{off}$ and $k_{on}$ are the kinetic off and on rates of binding, respectively. It is our observation that improvements in affinity during hit optimisation are usually through a slower $k_{off}$ or off-rate, as observed and exploited by others[22]. This rate is independent of concentration, and as we confirm in this paper, is a valid surrogate for measurement of affinity.

The process of hit optimisation in a drug discovery project requires a significant amount of time and resource in compound synthesis and purification followed by preparation of a solution of known concentration to assay for compound activity. This is particularly onerous in the early stages of discovery where many compounds may be synthesised to identify how to improve the affinity of binding by variation of substituents. We realised that the concentration independence of the off-rate could be exploited to speed up the efficiency and speed of compound optimisation. Changes in off-rate can be measured by SPR for a mixture where a starting material has been reacted with a second reagent without purification (so-called crude reaction mixture (CRM)) if the reaction has generated some product. This approach can be readily scaled to plate-based format, where the starting material is reacted with a different second reagent in each well of the plate. Provided there is no interference from the other reagents in the reaction, this significantly increases the speed, reduces chemical waste and lowers the cost of initial SAR exploration.

In this paper, we investigate how high-throughput crystal structure determination on the XChem platform (controlled by XChemExplorer, XCE[23]) can be used with CRMs and combined with SPR in hit optimisation. The results of experiments are described for two targets, the N-terminal ATPase domain of heat shock protein 90α (aa9–236), hereafter referred to as Hsp90 and pyruvate dehydrogenase kinase 2 (aa16-407), hereafter referred to as PDHK2, both members of the GHKL family of ATPases for which we have previously identified a series of potent inhibitors[13,14,24,25]. The initial experiments establish optimal soaking conditions for crystal structure determination. We then analyse the diffraction data for crystals of both PDHK2 and Hsp90 soaked with CRMs, and compare the results to the off-rates determined by SPR. The results suggest a protocol for such hit optimisation and identify issues to be considered in experiment design.

## Results

**Crude reaction mixtures.** Two starting materials were selected for the experiments reported here: **1** is a resorcinol fragment hit (SM1) and **2** (SM2), a hit compound evolved from **1**. These were chosen as representing hits with relatively fast (**1**) and slow (**2**) $k_{off}$ rates: **1** has a fast $k_{off}$ at the limit of sensitivity of SPR of more than $1\ s^{-1}$ for both targets; **2** has a slower $k_{off}$ for PDHK2 ($0.083\ s^{-1}$) than for Hsp90 ($0.2\ s^{-1}$).

Figure 1 shows the crystal structure of SM1 and SM2 bound to Hsp90 and PDHK2. For both proteins, the main interaction of the resorcinol core is with a solvent network around the acid moiety (D93 for Hsp90; D282 for PDHK2), and there are many similarities in the nature and geometry of the amino acids that interact with this core. The nitrogen of the SM1 amide provides a vector into solvent where the amino acids are different between the two proteins. It is this vector for which the CRMs were designed to explore in a fragment-growing approach, maintaining the interactions made by the resorcinol moiety in both SM1 and SM2.

Eighty-three different CRMs were prepared as described in our earlier publication[20] with the reaction scheme as summarised in Fig. 2 and in Supplementary Methods.

The CRMs were constructed through synthesis of the relevant two acid chloride intermediates with a protected resorcinol, which were then coupled with a selection of amines to give 55 and 28 CRMs with SM1 and SM2 as the starting material, respectively. Supplementary Table 1 contains the chemical structures for the starting material and expected product for all the CRMs used in

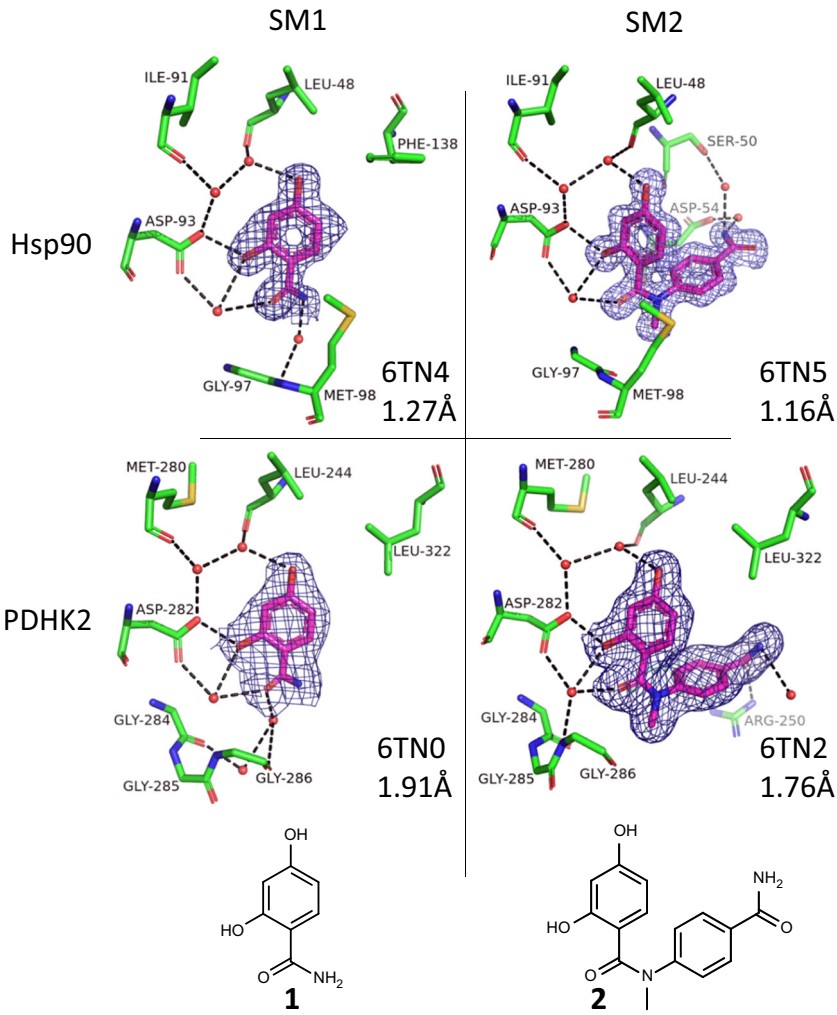

**Fig. 1 Crystal structures of the starting materials bound to the two proteins.** Shown are the compounds (magenta carbon atoms) and water molecules and amino acids from the proteins that are within 3.5 Å of the compound (with green carbon atoms) and with red oxygen, blue nitrogen and yellow sulfur atoms. 2Fo–Fc electron density contoured at 1σ for the ligand is shown in dark blue (see Supplementary Fig. 6 for omit maps); hydrogen bonding between ligand, protein and solvent is shown with black dashed lines; pdb code and resolution shown for each structure.

these experiments. The yields of product in the CRMs were determined by LCMS. This table also includes the results of analyses of crystal diffraction data from these samples soaked into Hsp90 or PDHK2 apo crystals that will be described and discussed below.

The expectation was that soaking crystals with the CRMs would give a native structure, the respective starting material or the expected product of the reaction bound to the protein. In the following description of the experiments and results, the terms "SM" and "PR" signify that the electron density determined for a crystal soaked with a CRM was interpreted as having, respectively, the starting material or the product bound. "N" is for a native structure. These terms, SM and PR, are used extensively in the following presentation of "Results" and "Discussion".

**Establishing and validating crystal-soaking protocols.** The first set of experiments were to optimise the crystal preparation and soaking protocols for the two proteins. The XChem facility requires crystals in triple-drop mosquito sitting-drop plates[26]. Our previous projects on the two proteins used crystals grown in hanging drops. In preliminary experiments (data not shown),

PDHK2 crystals could be grown in sitting drops, diffracted and were suitable for compound soaking and withstood 10% DMSO in the crystallisation medium. However, Hsp90 crystals grown in sitting drops could not tolerate DMSO and did not diffract when soaked with compounds for unknown reasons. Hsp90 crystals were therefore generated in hanging drops and transferred manually to sitting drops for the XChem robotic systems.

The details of crystallisation, the XChem process and software protocols used for structure determination and refinement for both manual and automatic refinement with the PanDDA[12] system at XChem are described in the "Methods" section. The rationale for performing all crystal-soaking experiments in triplicate is described in Supplementary Note 1.

A series of experiments were performed varying soaking time and concentration of the compound for crystals of PDHK2 soaked with 13 CRMs (indicated in Supplementary Table 1) and the corresponding 13 purified products (compounds **1** and **11–22**, see Supplementary Fig. 1). Crystals were soaked in triplicate for 4 h, overnight and 2 days with a final concentration of 2 mM of compound (final concentration of 10% DMSO) before diffraction data were collected for the total of 234 crystals on the XChem pipeline, and the diffraction data processed manually. The criteria for success were obtaining electron density for a

**Fig. 2 Reaction scheme for 55 and 28 crude reaction mixtures (CRMs) generated from SM1 and SM2.** Reagents and conditions: a Ac$_2$O, c H$_2$SO$_4$, 65 °C, 30 min, 80%; b i. (COCl)$_2$, DMF (cat.), DCM, rt 2 h; ii. R$^1$R$^2$NH, Et$_3$N, DCM, rt, 26 h; iii. NH$_3$ (7 N in MeOH, rt, 48 h); c BnBr, K$_2$CO$_3$, DMF, rt, 1 h; d i. **4**, (COCl)$_2$, DMF (cat.), rt, 1.5 h; ii. **6**, Et$_3$N, DCM, 0 °C → rt, 1.5 h, 94% (from **5**); e H$_2$, 10% Pd/C, EtOAc, rt, 2 h, 66%.

compound (either product or starting material in the case of the CRM soaks) bound to the protein. For a 2-day soak at 2 mM, 10% of the 234 datasets were unusable due to system failure or poorly diffracting crystals—there was little variation in failure rate with 4 h and overnight soaking (8% and 13%, respectively). A compound was identified as binding for 95%, 92% and 94% of the usable datasets for 4 h, overnight and 2-day soaks, respectively, for the CRMs, and 96%, 94% and 94% for the purified compounds. There was a slight degradation in resolution by using increased soaking times (see Supplementary Fig. 2), but the electron density for some compounds was stronger for the longer-time soak (an example is shown in Supplementary Fig. 3 for both a CRM and a purified compound). A 2-day soak time was therefore used in all subsequent experiments. The results with 2 mM compound in the soaking experiment were compared with 1 mM and 0.5 mM final compound concentration, all after a 2-day soak (data not shown). There was no improvement in resolution with 1 mM final compound concentration, and there were fewer bound structures when the crystals were soaked at 0.5 mM of CRM.

This soaking protocol (2 mM for 2 days) had been successfully used for Hsp90 crystals previously, so a smaller number of control experiments were carried out to confirm that this protocol was suitable for Hsp90 in the XChem system. Data were collected on single samples for six of the CRMs (as marked in Supplementary Table 1) and two of the expected purified compounds. Compounds were observed in all the datasets collected. The 2 mM for 2-day soaking protocol was used for both proteins for all subsequent experiments.

The datasets from the control experiments were also processed automatically using the PanDDA software[12] (see "Methods" for protocols). PanDDA identifies a binding event by comparison of an electron-density map with an average map built from a selection of the datasets provided. This is successful for fragment screening, as the hit rate (a fragment binding to a particular site of the protein) is low (0–5%), and so the average map built by PanDDA represents an empty binding site. For our experiments, the hit rate increases, which results in more density in the binding

site for the average map, which means PanDDA is less able to correctly identify a binding event. A modified protocol was used for the full screen, where 40 datasets were provided from crystals soaked with the screen solvent, DMSO. These "blank" datasets were used to build the PanDDA mean map, improving the contrast with the maps obtained from crystals with soaked compounds.

In the case of PDHK2, there was also an issue with the resolution of some of the data sets that were at the limit of where PanDDA can be effective (around 3.0 Å). In addition, the elaboration from SM2 can result in a disordered portion of the compound out into solvent (see, e.g., the electron-density maps shown in Supplementary Fig. 4). For Hsp90, issues also arose because different protein conformations can be found for different ligands[27]. Improvements were made in the protocols for using PanDDA (described in "Methods"), and all the results presented in this paper from PanDDA are with this improved protocol.

Having established the soaking conditions, the full set of 83 CRMs were soaked in triplicate into crystals of PDHK2 and Hsp90 at 2 mM for 2 days, and diffraction data collected on all 498 (83 × 3 × 2) crystals in a single dataset-collection session. The results are summarised in Supplementary Table 1 and discussed below.

**XChem performance and comparing PanDDA with manual fit of electron density**. When establishing the soaking protocol (described in the previous section), we had determined the structure of the purified compound in PDHK2 for 13 products from 83 CRMs. For the corresponding CRM, PR was seen for 11 of the compounds and SM for 2—although these were both CRMs with low product yield (<10%). From this, we conclude that soaking with a CRM finds most of the PR structures for PDHK2. For Hsp90, there were only two purified compounds for which structures had previously been determined (as marked in Supplementary Table 1), and PR was seen for only one of these from the corresponding CRM.

**Table 1 Fitting of electron density by PanDDA or manually gives similar results.**

| | No of PR agrees | PanDDA more PR than manual | PanDDA less PR than manual | No of SM agrees | No with >0 PR PanDDA | No with >0 PR manual |
|---|---|---|---|---|---|---|
| PDHK2 | 62 | 6 | 15 | 63 | 55 | 53 |
| Hsp90 | 68 | 15 | 0 | 63 | 36 | 29 |

Comparison of the results from PanDDA and manual fitting of diffraction data from the different CRMs soaked into both PDHK2 and Hsp90 crystals.

The electron-density maps generated for the triplicate soaks of 83 CRMs into PDHK2 and Hsp90 crystals were analysed manually and using the automated protocols of the PanDDA software. These results are summarised in the central columns of Supplementary Table 1 (columns 8–13 and 15–20). Each CRM contains varying amounts of product or starting material, depending on the degree of completion of the reaction, but the combined amount of product and starting material is high and approximately the same for each CRM, so if crystal soaking is successful, then a compound should be seen bound in every crystal.

The first feature of the results is that out of 498 different experiments, only 11 N was seen for PDHK2 and 43 N for Hsp90. There were only four CRM triplicates for Hsp90 (and none for PDHK2) where only N was seen—all others had at least 1 PR or 1 SM. This confirms that the soaking protocol chosen for both proteins was appropriate for the full set of CRMs, giving a success rate (compound bound) much higher than historically obtained (Supplementary Note 1) for the targets, probably because the starting materials consistently gave bound structures.

PanDDA and manual fitting find at least 1 PR in 50 of 83 CRMs for PDHK2 and 29 of 83 CRMs for Hsp90. The consistency of the results between PanDDA and manual fitting is illustrated in Table 1, which summarises for how many CRMs the number of PR or SM seen for either target is found using either PanDDA or manual fitting method.

These results illustrate firstly that success was maintained in the soaking and data collection for this larger set of samples, and secondly that PanDDA (after the improvements described in "Methods") and manual fitting find approximately the same number of PR and SM. The main discrepancy is the number of PR for PDHK2—which as discussed in the section establishing the protocols (and shown in Supplementary Fig. 4) is mainly due to low resolution of some of the datasets and disordered density as the compound is extended into solvent. This effect was compounded by the choice of dataset submitted to PanDDA by the automated pipeline XCE[23]. PR was fitted more often when the electron-density maps were fitted and refined manually.

**Treatment of the results from triplicate experiments**. The results from the triplicate soaking experiments were often not consistent. For example, although there were 63 CRMs where both PanDDA and manual find the same number of SMs for PDHK2, there were only 19 samples where the triplicate gave 3 SMs from PanDDA and 22 samples where the triplicate gave 3 SMs from manual fitting. As the aim of the experiments is to assess the value of the automated protocols of XCEM at identifying opportunities for hit optimisation, the results were treated as follows:

- The results from PanDDA fitting were used alone.
- If the electron density is poor or there was mechanical or crystal failure, then the result is N.
- Where there is at least 1 PR in the triplicate, then the

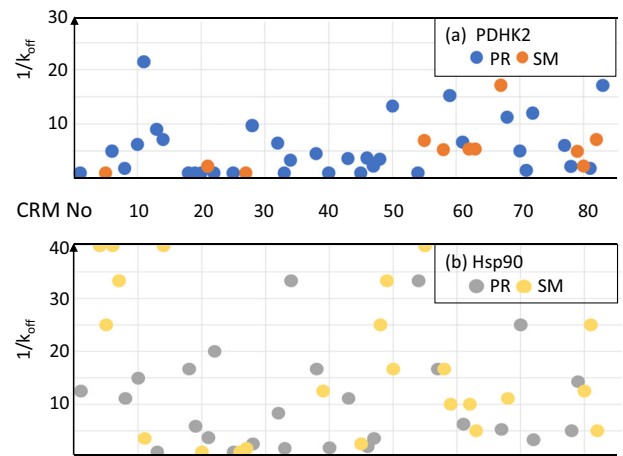

**Fig. 3 The XChem result does not correlate with affinity of the product.** $1/k_{off}$ (where determined and capped at 40 in the plot) for those crude reaction mixtures (CRMs) that gave a product in yield > 10% with a PR or SM for each of (**a**) PDHK2 (blue PR, $1/k_{off} = 5.0 +/-4.9$; orange SM, $1/k_{off} = 5.4 +/-4.7$) and **b** Hsp90 (grey PR, $1/k_{off} = 16.4 +/-16.6$; orange SM, $1/k_{off} = 25.1 +/-39.8$). Values are mean $+/-$ standard deviation. Statistical analysis in Supplementary Fig. 5a.

result is PR.
- Where there is no PR but at least 1 SM, the result is SM.
- Otherwise the result is N.

All reference to PR, SM and N in the following analysis is from this treatment.

**Off-rate for the CRMs**. The off-rate or $k_{off}$ for dissociation of each of the 83 CRMs from Hsp90 and PDHK2 was measured in duplicate by SPR and is shown in Supplementary Table 1. Failure to obtain a $k_{off}$ value for some CRMs (not determined, ND) was due to poor signal and was most marked for PDHK2.

The first analysis is to determine whether the XChem result can provide information in the absence of affinity data. Figure 3 shows a plot of $1/k_{off}$ (proportional to affinity) for those CRMs with a yield above 10% that gave either an SM or PR structure for the two targets.

The plots in Fig. 3 emphasise three points. The first is that the compounds exhibit a slower $k_{off}$ and thus higher affinity for Hsp90 than for PDHK2. The second is that PDHK2 gives PR more often than SM. The final observation is that SM is obtained for many of the slower $k_{off}$ compounds for Hsp90 and PR for many of the faster $k_{off}$ compounds for PDHK2, so there would be many false positives and false negatives if only the crystal structure was used to guide compound optimisation. As shown in Supplementary Fig. 5b, there is a suggestion that for Hsp90, the crystals are selecting starting material for products with more rotatable bonds, even though the product has a slower $k_{off}$, but this is not a significant effect.

| Table 2 Analysis of results. | | | | | |
|---|---|---|---|---|---|
| | | PDHK2[d] | | Hsp90[e] | |
| | $k_{off}$ | No PR | No SM | No PR | No SM |
| CRM from SM1[a] | Faster[c] | 16 | 5 | 2 | 1 |
| | Slower | 19 | 3 | 26 | 22 |
| CRM from SM2[b] | Faster[c] | 9 | 14 | 3 | 4 |
| | Slower | 2 | 2 | 7 | 14 |

The table shows the number of PR and SM for each target for the CRMs from the two starting materials where the product gave a faster (or equal) or slower $k_{off}$ than the starting material.
[a]There are 55 CRMs from SM1; for Hsp90, two gave native structures.
[b]There are 28 CRMs from SM2.
[c]Faster or equal to $k_{off}$ for the starting material.
[d]$k_{off}$ was not determined for 12 CRMs from SM1 and 1 CRM from SM2 for PDHK2.
[e]$k_{off}$ was not determined for two CRMs from SM1 for Hsp90.

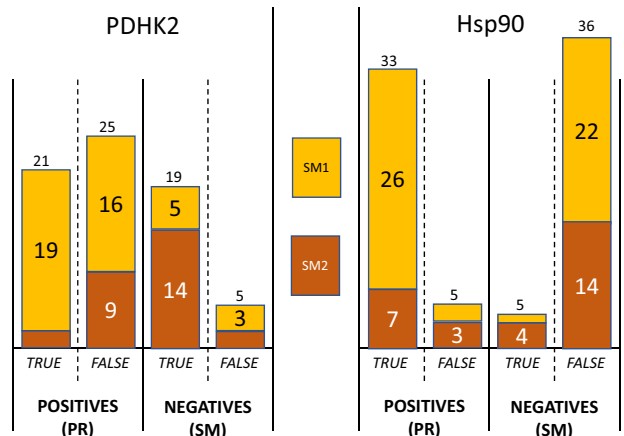

**Fig. 4 Analysis of the results.** The data of Table 2 plotted to emphasise the number of true and false positives and true and false negatives obtained.

A more complete analysis of the results of Supplementary Table 1 is to separately consider the CRMs for the two different starting materials, as SM1 has a faster $k_{off}$ than SM2. Table 2 summarises the results, separated on starting material and the measured $k_{off}$ for the two targets. If the $k_{off}$ for the product in the CRM is slower (higher affinity) than the starting material, then PR is a true positive and SM a false negative; if the $k_{off}$ for the product in the CRM is faster or equal (weaker affinity) than the starting material, then SM is a true negative and PR a false positive. Figure 4 reorganises the results into true and false positives and true and false negatives.

The cases where there are five or fewer examples are not discussed further—this rate of false positives and false negatives is to be expected in a high-throughput experiment. Figure 4 emphasises two issues—the high number of false positives seen for PDHK2 and the high number of false negatives seen for Hsp90. These are analysed further.

For PDHK2—16 of the 21 CRMs generating a product with $k_{off}$ equal to or faster than for SM1 gave PR and only five gave SM. This is not as expected as the product is as weak or weaker at binding to PDHK2 than the starting material, and PDHK2 was in general more reliable for crystal soaking. This does not appear to be related to yield as 7 of these 16 CRMs that give PR have yields below 20%, so there should be enough SM1 available. A possible explanation for some of these results is that SM1 has a much weaker binding than the products in these cases. The $k_{off}$

measurements are at the limits of sensitivity of SPR, and any $k_{off}$ faster than $1\,s^{-1}$ was set to $1\,s^{-1}$. However, if the CRMs are also profiled by SPR, such false positives can readily be identified.

For Hsp90—14 of the 21 CRMs generating a product with slower $k_{off}$ than SM2 gave a structure of the starting material. This is not as expected as the product is binding with higher affinity than the starting material. The yield was >40% for all seven of the cases where PR was seen. Eight of the 14 cases where SM was seen had product yield <10%. The products for the other six CRMs with yield >10% that gave SM have a predicted solubility of close to zero. The number of false negatives from the CRMs from SM2 could therefore be due to either low yield or poor solubility of the product formed.

The other anomaly for Hsp90 cannot be fully explained. Twenty-two of 48 CRMs generating a product with slower $k_{off}$ than SM1 gave SM. This is not as expected as the product binds with higher affinity than the starting material. This could either be related to yield as this was less than 10% for 9 of the 22 CRMs or it could be due to low solubility—the product for 9 of the 11 remaining CRMs with yield above 10% has predicted solubility of less than 1 mM. Therefore, the false negatives in this case can be explained by solubility or yield. However, low solubility or yield is not always a predictor of whether SM is observed instead of PR. Eight of the 26 CRMs that gave a PR also had less than 10% yield, and the product for 15 of these 26 CRMs has predicted solubility of less than 1 mM. However, it should be noted that the determination of yield by automated LCMS is not fully reliable, particularly for compounds with poor solubility.

There is one example that illustrates the influence of product yield in the CRM. CRM numbers 32 and 44 are duplicates—however, the yield for CRM 32 was much higher (43%) than that for CRM 44 (2%). This can explain the observed kinetics—for CRM 44, the $k_{off}$ is that of the starting material. The amount of product is also reflected in the number of PR and SM seen. All three of the triplicate crystals for CRM 32 gave PR for PDHK2, and two of the triplicates gave PR for Hsp90, whereas only one of the crystals for CRM 44 gave PR for PDHK2 and none for Hsp90.

**Is there an explanation for the significant anomaly for Hsp90 samples?** The above analysis highlights that Hsp90 gave many false negative results. This is where SM is seen when it should have been PR from CRMs that give a good yield of a product with a slower $k_{off}$ than the starting material. There are four possible reasons why these compounds did not give PR—solubility of the product in the crystallisation conditions, there was an error in measuring $k_{off}$ by SPR, the product does not soak into native Hsp90 crystals or the CRM had some influence on product binding to Hsp90 in the crystal. Purified compounds were available for some of these CRMs, and for this subset, there are two cases. The first is where PR is seen for PDHK2 from the same CRM, suggesting that it is not an issue of solubility; this is for CRM numbers 2, 11, 15 and 63. The second case is where SM is seen for PDHK2 (mostly explained by $k_{off}$ being faster than the starting material for PDHK2), but SM is also seen for Hsp90 where the $k_{off}$ is slower than the starting material. This is for CRMs 57, 68, 72, 75 and 83. CRMs 2, 11 and 15 are from SM1; the remainder are from the relatively slower $k_{off}$ SM2.

Table 3 summarises for each of the purified compounds from these nine CRMs, the results of a manual soaking of the purified compound or the CRM into Hsp90 crystals, and the measurement for the purified compounds of solubility and $K_D$ measured by both SPR titration and by isothermal titration calorimetry (ITC). A crystal structure of product was seen for eight of the nine purified compounds (not for the product of CRM 11 which had low solubility), and one of the CRMs gave PR on manual soaking

**Table 3 Binding of ligands from crude reaction mixtures (CRMs) in Hsp90 crystals is not related to affinity.**

| CRM no | Solubility of pure (µM) | Result soak of pure | Result soak of CRM | $k_{off}$ CRM ($s^{-1}$) | Yield (%) | HAC | nRot | $K_D$ (SPR) (µM) of pure | $K_D$ (ITC) (µM) of pure |
|---|---|---|---|---|---|---|---|---|---|
| 2 | 400 | Bound | PR | 0.1 | 8 | 22 | 1 | 1.0 | 0.65 |
| 11 | 24 | Native | SM1 | 0.23 | 34 | 30 | 4 | 17 | ND |
| 15 | 440 | Bound | SM1 | 0.06 | 1 | 20 | 3 | 0.07 | 0.36 |
| 63 | 460 | Bound | SM2 | 0.012 | 9 | 33 | 4 | 0.04 | 0.07 |
| 57 | 320 | Bound | SM2 | 0.03 | 39 | 32 | 3 | 0.71 | 0.26 |
| 68 | 350 | Bound | SM2 | 0.12 | 9 | 29 | 4 | 3.0 | ND |
| 72 | 300 | Bound | SM2 | 0.12 | 8 | 30 | 6 | 1.7 | 3.3 |
| 75 | 7 | Bound | SM2 | 0.013 | 3 | 30 | 3 | 0.019 | ND |
| 83 | 370 | Bound | SM2 | 0.05 | 8 | 31 | 6 | 0.36 | 0.29 |
| SM1 | | | | >1 | | | | 2000 | |
| SM2 | | | | 0.2 | | | | 2 | |

The results for measuring solubility at pH 7.4 and affinity and manual crystal structure determination for selected CRMs and associated purified products. Solubility, SPR and ITC measurements were made at 25 °C. SPR measurements are an average of at least two determinations; ITC and solubility are a single measurement. ND in ITC was due to lack of heat observed.

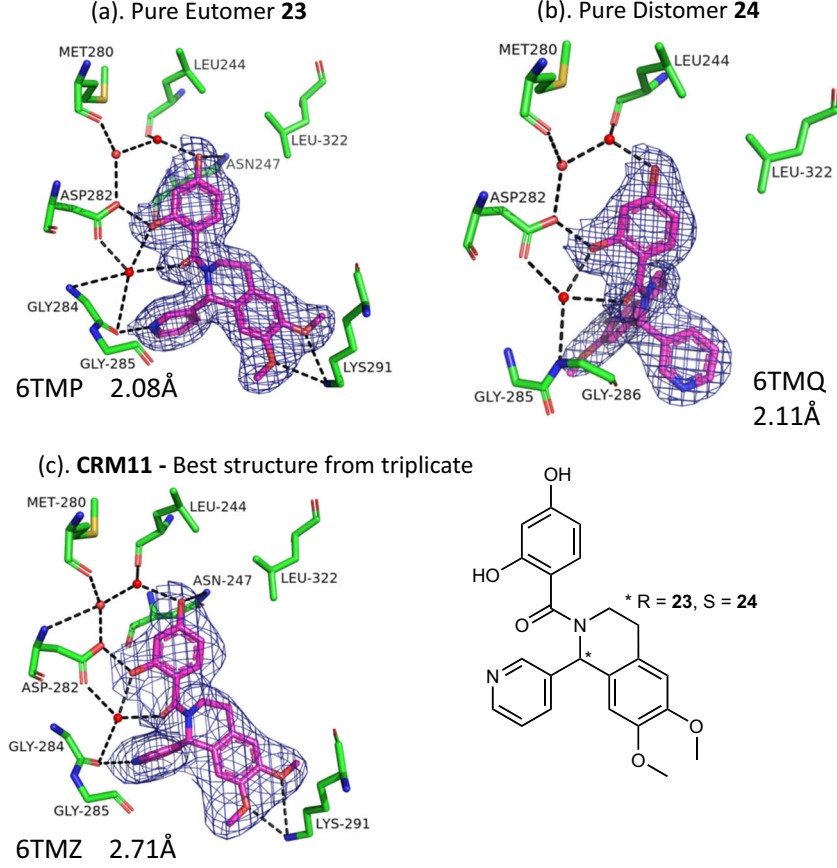

(a). Pure Eutomer **23**

6TMP   2.08Å

(b). Pure Distomer **24**

6TMQ 2.11Å

(c). **CRM11** - Best structure from triplicate

6TMZ   2.71Å

* R = **23**, S = **24**

**Fig. 5 Crystal selects the most potent enantiomer from a mixture.** Details from the crystal structures of **a** eutomer **23**, **b** distomer **24** and **c** crude reaction mixture (CRM) number 11 bound to PDHK2. PDB code and resolution are shown. See the legend to Fig. 1 for details of atom selection and representation.

(CRM 2). This leaves anomalies for seven CRMs that gave SM. One of these had low solubility (CRM 75), leaving six CRMs giving SM on repeated manual soaking to explain.

For two of the CRMs (68 and 72), the more accurate measurement of $K_D$ by titration in SPR shows that the product has about the same affinity as the starting material. The $k_{off}$ values measured in high throughput for the CRMs were similar to the starting material (0.12 $s^{-1}$ for both CRMs versus 0.2 $s^{-1}$ for SM2), suggesting hat a larger threshold should be used to define a slower $k_{off}$. Of the remaining four (CRMs 15, 63, 57 and 83), three had <10% yield of product, which may be the explanation why SM was

obtained. Taking into account the contributions of solubility and product yield to the likelihood of obtaining PR, the results are reasonably consistent for such a high-throughput experiment.

**Discriminating between eutomer and distomer.** CRM number 11 is a racemate of two enantiomers, **23** and **24**. In a series of control experiments, purified samples of each of the separate enantiomers were soaked (2 days, 2 mM) into PDHK2 crystals, and as shown in Fig. 5, structures were obtained for both the eutomer (**23**, $K_d = 0.14$ µM determined by FP assay) and distomer (**24**, $K_d = 17$ µM). A PR was obtained for all three of the

**Table 4 Results obtained if we consider the singleton or duplicate experiment compared to the triplicate.**

| | PDHK2 | | | HSP90 | |
|---|---|---|---|---|---|
| Triplicate result class | Number from triplicate | Number from singleton (duplicate) | Singleton/duplicate result class | Number from singleton (duplicate) | Number from triplicate |
| N | 0 | | | | 4 |
| SM | 28 | 6 (2) | N | 6 (1) | 43 |
| | | 22 (26) | SM | 37 (42) | |
| PR | 55 | 13 (5) | N | 5 (2) | 36 |
| | | 2 (6) | SM | 1 (1) | |
| | | 34 (44) | PR | 30 (33) | |

soaks of CRM number 11 in the full XChem screen that selected just the higher-affinity eutomer **23**.

## Discussion

The screening of CRMs can increase the efficiency of early hit exploration in drug discovery. The time-saving and reagent efficiency will vary with reaction chemistry, but the reactions reported here for 83 CRMs took ~5 FTE days to set up and analyse, compared to an estimate of 15–20 FTE days for preparation, purification, characterisation and preparation of separate stock solution for each of the samples, additionally saving 35–50 litres of various solvents for workup and purification, and approximately sevenfold saving of reagents and solvents for synthetic operations (detailed calculation in Supplementary Note 2).

The objective of this study was to investigate how the streamlined sample handling, data collection and processing of the high-throughput structure determination facility, XChem at the Diamond Light Source, UK synchrotron, can be used to screen such CRMs containing compounds that explore how to grow from one vector of a fragment hit to a more potent lead compound. As with any high-throughput experimental approach, there are false positives and false negatives. The data collected were able to identify these and the reasons for discrepancies explored.

The optimal conditions for obtaining soaked crystal structures of compounds bound to the two test proteins, Hsp90 and PDHK2, were established in control experiments. This also established that the automated processing by the PanDDA software (with suitable modifications to the protocol) was as reliable as manual fitting at identifying the bound compound in the electron density. Most of the analysis therefore focussed on the results from PanDDA as it enables the rapid analysis of a large number of crystallographic datasets.

The experiments reported here were performed in triplicate, and as described above, the result classified as the product bound to the protein (PR) if any of the triplicates gave PR, as the starting material bound to the protein (SM) if there was no PR in the triplicates but an SM, and native (N) if no result was obtained for any of the triplicates. The main objective of soaking the CRM into a crystal is to obtain the structure of a product bound to the protein to inform structure-based compound optimisation. Table 4 summarises the different results obtained if just the first (or first two) experiment(s) of the triplicate is considered. For PDHK2, taking just the first result would have given 34 PR instead of 55, and for Hsp90, 30 PR instead of 36. For the other 15 CRMs for PDHK2 (6 for Hsp90), the first experiment was recorded sometimes as SM, but predominantly as N because of some failure of the crystal, data collection or processing; for these cases, the duplicate would have improved the results. It is therefore a balance of resources available compared to the value

of completeness of the data obtained in deciding how many repeats to perform for each experiment.

The central premise of assessing CRMs by SPR is that the measured off-rate, $k_{off}$, is a surrogate for affinity, such that slower $k_{off}$ reflects higher affinity. This is confirmed for some examples in the results shown in Table 3 above. SM1 has a $K_D$ of 2 mM measured by ITC for binding to Hsp90 with a very fast off-rate ($k_{off} > 1\,s^{-1}$) by SPR. The purified compounds from the CRMs 2, 11, 15 and 55 all have slower $k_{off}$ by SPR and higher affinity as measured by ITC.

The CRMs contain both starting material and product, and the expectation was that the product would be bound where it had a higher affinity and the starting material where the modification reduced the affinity. One clear result from the experiments was that the crystal can select the highest-affinity enantiomer from a racemic mixture within the CRM, which confirms this principle. In about a half of the examples (Table 2), the results were as expected. There were indications that in other examples, the product yield may be the issue. For example, there were just four examples where only native structures were obtained for Hsp90 from 83 CRMs (see Supplementary Table 1, CRM numbers 16, 26, 45 and 51). Two of these four CRMs contained no detectable product, but did contain starting material (starting material was also seen for these two CRMs for PDHK2). The other two CRMs contained a good yield of product, and a product structure was seen in PDHK2 suggesting good solubility, but nothing was seen bound in Hsp90. More confounding was that for some CRMs, a product-bound structure was observed even when the amount of product generated was not detectable in the high-throughput LCMS measurements, and in other cases, the product was seen when SPR indicated that the product had a weaker affinity for the protein than the starting material. The analysis was able to identify a reason for some but not all these anomalies. Another consideration is that the compounds could have different solubility in the crystallisation systems, which is different from that of the buffers used for SPR. It could also be that the kinetics and/or thermodynamics of binding to a preformed crystal is affected by the crystal packing in different ways for different compounds. There is a suggestion (Supplementary Fig. 5b) that SM was seen for larger products with a slower $k_{off}$; however, the results with purified compounds (Table 3) suggest that this is only when the compound is in a CRM.

In conclusion, soaking a protein crystal with a CRM automatically generated crystal structures for most (for PDHK2) or many (for Hsp90) of the compounds generated in the reaction that have an increased affinity for binding to the protein compared to the starting material. For PDHK2, most of the false positives and false negatives can be understood to arise from low solubility or low yield of product compound. There was a higher false-negative rate for Hsp90 where crystal structures were of the starting material rather than the higher-affinity product. The explanation for this anomaly is not clear, but is possibly related to

the size of the compounds that would not readily soak into crystals in the conditions under which the XChem pipeline was operated.

There are some obvious limitations in applying the lessons learnt from this study to other optimisation projects. The results are for two enzymes with similar active sites (though very different activities) and for a focussed class of compound. It may be that other proteins, other compound classes (and optimisation strategies) and reaction types will behave differently. However, this exhaustive analysis suggests the following protocol:

1. Establish a suitable crystal form (and format) for soaking with a high concentration of ligand in the XChem facility. There can be surprising differences in the performance of what appear to be identical crystals grown in different formats (in the Hsp90 case sitting versus hanging drops).

2. Characterise how reproducibly the crystal system will generate ligand-bound crystal structures using the XChem protocols and PanDDA analysis system. Use several known compounds of varying affinity and molecular dimensions.

3. Use XChem (or some other screening method followed by crystal structure determination) to identify fragments that bind, and which vectors are accessible (synthetically and as revealed by the binding mode) for optimisation of the fragment; if possible, use SPR or NMR to characterise binding of the fragment and evolved fragments.

4. Construct CRM libraries from selected fragments (consider the strength of binding of the starting material and likely solubility of the product); measure the yield of the product in each CRM by mass spectrometry.

5. Use XChem to determine the structures of crystals soaked with CRMs. It is probably enough to perform these screening experiments as singletons. If possible, normalise the concentration of the CRM to compensate for any low yield of the product.

6. Use SPR to characterise the $k_{off}$ of the CRMs.

7. Assess whether enough information for compound products is obtained that can guide hits to lead optimisation. If XChem has not generated product-bound structures for sufficient products where the measured $k_{off}$ from SPR is slower, then the crystal soaking should be repeated for selected CRMs, and if still not successful, consider re-synthesis (purification) of the compound.

Using such a protocol will more rapidly generate structural insights on which to design modifications to compounds to improve affinity or other drug-like properties, thereby increasing the speed and efficiency of the drug discovery process.

## Methods
**Protein production**. Both proteins (human PDHK2 (residues 16–407) and N-terminal domain of human Hsp90α (residues 1–236)) were expressed as an N-terminal His tag fusion protein with a TEV protease cleavage in BL21 cells and purified on a $Ni^{2+}$ affinity followed by size-exclusion column, details as previously described[13,27]. Uncleaved His-tagged variants were used in SPR experiments; tags were removed for crystallisation.

**Crystallisation**. PDHK2 was mixed with the allosteric binding site compound Pfz3 (N-(2-aminoethyl)-2-(3-chloro-4-[(4-isopropylbenzyl) oxy]phenyl) acetamide); Pfizer, Patent application: EP1247860, 1-304, 2002) at threefold molar excess before being concentrated to 10 mg/ml. Crystallisation experiments were set up in sitting-drop 96-well 3-drop Swissci plates (Molecular Dimensions Limited) using a Mosquito crystal robot (TTP Labtech Ltd). The liganded Pdhk2 crystallised in 0.1 M sodium acetate, pH 5.8, and 0.125 M $MgCl_2$ at 4 °C within 48 h.

Hsp90, concentrated to 18.75 mg/ml, was crystallised in 10% PEG3350, 0.1 M sodium cacodylate, pH 6.5, and 0.2 M $MgCl_2$ using the hanging-drop vapour-diffusion method at 4 °C. Again, crystals grew within 48 h. The crystals were subsequently transferred into the correct format needed for the XChem robotic systems by pipetting 500 nL of mother liquor containing at least one Hsp90 crystal

into each depression of a 96-well 3-drop Swissci plate. This was repeated until >300 Hsp90 crystal drops were available.

**Crystal structure determination**. All methodology details of the XChem fragment-screening platform can be found on the XChem webpage that is accessible from the Diamond Light Source homepage (https://www.diamond.ac.uk/Instruments/Mx/Fragment-Screening/Methods-for-Fragment-Screening.html). Details of individual parts of the platform have been published[12,23,26].

The crystal drops for both proteins were first photographed using a RI1000 Formulatrix imager. Each image was ranked according to the presence and quality of crystals by the TeXRank software. An ECHO acoustic liquid handler (Labcyte) was used to transfer the CRMs/pure compounds to the crystal drops, aiming for an area of the drop that would cause minimum disruption to the crystal, as determined by visual inspection following the TeXRank step.

**Control experiments for PDHK2 crystal structure determination**. A final concentration of 2 mM of each of the 13 CRMs and 17 pure compounds was added into crystal drops of PDHK2 in triplicate for soak-time durations of 2–4 h, O/N and 2 days at 4 °C. In addition, the final compound concentrations of 0.5 mM and 1 mM were added into PDHK2 crystal drops in triplicate, which were subsequently left to soak for 2 days also at 4 °C. Glycerol was added as a cryoprotectant (at a final concentration of 20%) by ECHO acoustic transfer shortly before mounting the crystals in cryo-cooled pucks. All data were collected on beamline I04-1 in auto-mated mode and autoprocessed using xia2[23].

**Control experiments for Hsp90 crystal structure determination**. The same XChem equipment was used for the Hsp90 control experiments. However only six of the PDHK2 CRMs and the equivalent purified compounds were added into Hsp90 crystal drops at 2 mM for 2 days at 4 °C. The cryoprotectant (13% glycerol) was added, and the crystals mounted, cryo-cooled and data collected as described for the PDHK2-soaked crystals above.

**Collecting diffraction data for soaks of 83 CRMs into PDHK2 and Hsp90 crystals**. Following data analysis of the control experiments, the full set of 83 CRMs were soaked in triplicate into crystals of PDHK2 and Hsp90 at 2 mM for 2 days at 4 °C. The relevant cryoprotectants were added, crystals mounted and cryo-cooled and data were collected as described above for the control experiments.

**PanDDA analysis and improvements made to the PanDDA protocols**. The datasets were processed using PanDDA in XChemExplorer[23]. Structure solution by molecular replacement was performed using previously determined apo structures (Hsp90 $PDB_{code}$ 1uyl[27], PDHK2 $PDB_{code}$ 2bu7[28]) as the search models. The PanDDA system compares the electron density for each dataset against an average map to generate a so-called "event map". This is inspected manually and the appropriate ligand fitted; the system then continues to refine the structure and report fitting statistics.

The XChem facility is primarily used for fragment screening on one target. The average map used to identify ligand binding is built from all the datasets. This is effective for fragment screening in which the hit rate (a fragment binding in a particular site of the protein) is low (0–5%). In the experiments described here, the hit rate increases, and the quality of the average map built by PanDDA, if the settings are unchanged, becomes less sensitive as more datasets used to generate the average map have bound ligands. The comparison between maps derived from crystals with ligand bound and such a statistical model can miss binding events as the difference in electron density is poorly defined. As a workaround (and protocol for a follow-up screen), at least 43 (for Hsp90) and 88 (for PDHK2) datasets were collected that had only been soaked with the screen solvent (e.g., DMSO) using the same volume of solvent (e.g., 10% of the final total volume of the crystal drop). These datasets only are selected to build the PanDDA statistical model. The event maps obtained from soaking the follow-up compounds are much clearer, resulting in a competitive PanDDA analysis. This new protocol was used in this study for both proteins.

In the semi-automated PanDDA analysis, the event map is inspected manually to assign whether the dataset is native or is PR or SM. The ligand is then fitted manually, and the PanDDA process continues to automatically refine the structure, finally generating a 2Fo–Fc map that is passed to the CCP4 program EDSTATS[29]. Supplementary Table 1 shows the real-space correlation coefficient for the ligand on this map. Supplementary Note 3 also contains a description of preliminary calculations, which demonstrate that the real-space correlation coefficient value cannot be used automatically to assign SM or PR.

**PanDDA settings**. For PanDDA to build a statistical model allowing the detection of all binding events, some datasets were filtered using the following two options available on the PanDDA tab of XCE:

*ignore_datasets*: datasets that do not need to be analysed by PanDDA. Datasets from crystals obtained from a different crystallisation method, soaked using different parameters (crystal solvent and concentration), unsoaked controls. Leaving these datasets would either prevent the PanDDA analysis to succeed, or just reduce the quality of its output.

*exclude_from_characterisation*: Datasets analysed but excluded from the set that PanDDA can select to calculate the mean map and generate the statistical model. All the crystals soaked with CRMs were excluded. Including these datasets that have a high chance of containing a binding event would result in PanDDA generating an average map of low quality, and a poor analysis.

Although the published paper recommends the use of at least 30 datasets for the generation of a "complete" statistical model, more DMSO-only crystals were soaked to allow for experimental error and resolution variations.

**Manual analysis of crystallographic datasets**. Structure solution, refinement and initial electron-density map calculation for each of the datasets was carried out within the CCP4 programme suite[29], using scripts incorporating the molecular replacement programme Phaser[30] used to solve the structures, Refmac5[31] for refinement and COOT[32] for inspection of the electron-density maps. The data collection and refinement statistics for the crystal structures shown in Figs. 1 and 5 are shown in Supplementary Table 2. The omit maps shown in Supplementary Fig. 6 were calculated as $F_{obs}-F_{calc}$, where $F_{obs}$ was the observed structure factor for diffraction from the protein–ligand complex, and $F_{calc}$ the calculated structure factor for the refined model of the protein–ligand complex with the ligand and active-site solvent removed.

**Ligand fitting**. The programme Grade (Global Phasing Ltd) was used to create ligand PDB and geometry dictionary files in the PanDDA analysis, whereas AceDRG[33] was used in the manual analysis. In both, the PanDDA and manual analysis compounds were fitted into the electron-density maps using COOT[32].

**Surface plasmon resonance**. SPR experiments were conducted on T200 and T100 BIAcore instruments (GE Healthcare) at 25 °C. The His-tagged proteins were immobilised on a series of S NTA chips, and the running buffer with 10 mM HEPES, pH 7.4, 150 mM NaCl, 0.05% P-20, 0.025 mM EDTA and 1% DMSO was used. The sensor surface was regenerated between cycles to eliminate any carryover of protein and/or analyte. Details are as described previously[19].

**Isothermal titration calorimetry**. ITC experiments were conducted at 25 °C by titration of a stock sample of 100 μM compound onto a solution of 10 μM Hsp90 in a GE ITC200 instrument.

**Compound aqueous solubility**. Test samples for each compound (500 μM, 2.5% DMSO fc) were prepared in PBS (pH 7.4) and incubated overnight at room temperature on a plate shaker. Standard samples for each compound were prepared in 60:40 v/v methanol:PBS and also incubated overnight. All samples in PBS were filtered prior to measuring UV absorbance in the wavelength range 250–500 nm. Calibration lines were plotted for the standard samples for each compound using absorbance at the wavelength with the largest signal, and solubility (μM) for the test sample read from the calibration line. Data presented are the mean of three values determined within the same experiment.

**Statistical analysis**. Two-sample Kolmogorov–Smirnov tests[34,35] were performed in KNIME[36] using the "Kolmogorov–Smirnoff Test" node at alpha = 0.1 to test the null hypothesis that the SM and PR results for each target (HSP90 and PDHK2) were from a different distribution for the relevant property ($1/k_{off}$, HAC, nRot). Notched boxplots were generated in KNIME using the "Notched Boxplot" (JFreechart) node from the Vernalis KNIME Community contribution[37]. Notches are centred on the median and show the 95% confidence interval of the median ($+/- 1.57 \times IQR/sqrt(N)$). Notches may extend beyond the limits of the boxes for small values of N or skewed distributions.

## Data availability

PDB files containing refined coordinates of the structures presented in this paper are deposited at the RCSB. For Hsp90, the codes are **1**: 6TN4, **2**: 6TN5. For PDHK2, the codes are **1**: 6TN0, **2**: 6TN2, **23**: 6TMP, **24**: 6TMQ, CRM 11: 6TMZ. The authors will release the atomic coordinates and experimental data upon article publication.

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

## Acknowledgements

We thank Alan Robertson (Vernalis) for PDHK2 protein production and Julia Smith (Vernalis) and Margaux Ouvry (former Vernalis) for Hsp90 protein production. The SGC is a registered charity (number 1097737) that receives funds from AbbVie, Bayer Pharma AG, Boehringer Ingelheim, Canada Foundation for Innovation, Eshelman Institute for Innovation, Genome Canada, Innovative Medicines Initiative (EU/EFPIA) [ULTRA-DD Grant No. 115766], Janssen, Merck KGaA Darmstadt Germany, MSD, Novartis Pharma AG, Ontario Ministry of Economic Development and Innovation, Pfizer, São Paulo Research Foundation-FAPESP and Takeda and Wellcome [106169/ZZ14/Z].

## Author contributions

R.E.H., L.B. and J.M. planned the experiments; S.R. designed and performed all compound and CRM synthesis; N.M. performed all SPR measurements, J.M. performed all ITC measurements; A.S. generated all protein crystals; L.B. performed all crystallographic experiments and analysis, aided by A.A. and P.C. with input from F.v.D.; T.K. modified the PanDDa protocols; R.E.H. drafted the paper that was edited and approved by all authors.

## Competing interests

The authors declare no competing interests.
