## [Peer Review File · Communications Chemistry]

Reviewers' comments:

Reviewer #1 (Remarks to the Author):

While fragment-based drug discovery is a widely used alternative to high-throughput methods for initial lead identification, the development of fragments into high potent leads remains challenging. Here, the authors describe a method to efficiently screen for additional compounds with improved affinity by using crude reaction mixtures. This enables the detection of a vast number of additional binders in a material and time efficient manner.

I recommend publication after some minor corrections.

On page 7, line 136: Instead of 133 compounds it should be 143 compounds.

On page 9, Figure 1 and page 25, Figure 5: The electron density should be depicted as difference electron density (omit-map) at an appropriate level. At least the PDB codes should be given here and the refinement statistics should be referred to the suppl. material. Also, I assume that in Table S2 the complexes PDHK2 +3 and PDHK2 +4 refer to PDHK2 +23 and PDHK2 +24.

On page 9, Figure 1: I found it interesting that compound SM2 showed improved affinity, despite the fact that there seem to be no additional interactions of SM2 to the protein when compared to the SM1 complex. The SM2 additional part (when compared to SM1) seems to stick into the solvent. Maybe the authors can comment on that.

Page 27, last paragraph: Just out of curiosity, how does the hit rate improve if taken the results from only two experiments instead of three? Maybe 2 is the way to go?

Page 36: Author contributions: Include Frank von Delfts contribution to the paper.

Reviewer #2 (Remarks to the Author):

Referees report on: Rapid optimisation of fragments and hits to lead compounds from screening of crude reaction mixtures

The manuscript by Baker and colleagues describes important work to improve the 'hit to lead' process in the development of therapeutic or probe compounds. The authors use a 'crude reaction mix' to select compounds with higher affinity than the initial lead using the binding of the target protein to pull the compound out. The results are assessed using X-ray crystallography and surface plasmon resonance and are amenable to a high-throughput approach. While I believe this work presents an important and new method in compound development and merits publication in this journal; I found the manuscript to be very poorly written, confusing and difficult to read. I have made detailed comments below but for example the use of the acronyms PR and SM throughout I found very frustrating – I had to constantly refer back to what they were – why not just write product and starting material? These are just 2 examples. The tables and figures are poorly made and difficult to understand and need to be re-thought – there is a lot of information here and it could be presented in a much more intuitive fashion. A final small concern is that while the resolution obtained for the product bound complexes should be sufficiently high for unambiguous ligand determination the density in the reports looks often quite difficult to assign. I am not implying a lack of due diligence but I feel that this is an important aspect for the work presented here – identification of a ligand from a large mix – I feel the authors should describe at length how well they can ascribe the density to a specific component and how mistakes are avoided.

I have a number of general comments below that need answering and a large number of edits, that relate mainly to the use of language, that are found in 'Specific comments'.

Specific Comments:

Title: The title is a bit confusing – maybe remove 'and'

Abstract: line 29 insert 'the' before 'generation'

Line 31 replace enough with 'sufficient'

Line 34 and 35 – the reaction mixtures are not assessed but their binding to the target proteins, XChem has a remarkable reputation but I would say that high-throughput is sufficient rather than ultra-high and I would myself describe it as a data collection facility or a structure determination facility.

Introduction:

Line 52 is it OK to site a blog? If so I would prefer that the specific entry is cited – I could not find what is referenced here

Line 55 – large literature? A wide body of research?

Line 68 – prefer hundreds to thousands

Line 100 remove 'this'

Line 102 – remove 'ultra'

Line 106 and 107 replace hereafter with referred to as

It would be useful to expand on what exactly CRMs are here – they need to be properly introduced – maybe this is already well established but the legend for Table S1 seems to explain it so why not the intro? Also, The next section 'Historical data' is this still the introduction? Would be better in the results or methods. The introduction left me very unclear about what the point of the experiments were – the 2 paragraphs of the discussion expressed this very well – maybe move them here?

Results

Figure 1 Would be very good to have simulated annealing omit maps

Line 226 unknown reasons rather than the inverse.

Table S1 – there has obviously been a lot of effort put into this table but I found it so hard to interpret – for a start maybe better to separate the 2 proteins? Would be better to have these data in the main paper!

Line 259 – data were!

Line 271 – Do you mean by lower quality that because the active site contains a bound ligand you get a worse result to put all electron density on an absolute scale? I am not really sure why PanDDA is used here – it is excellent in being able to identify

Line 289 – single data set collection – do you mean a single experimental session?

Line 305 – would prefer column numbers or something more descriptive

Line 341 – products could be placed into manually assessed electron density maps?

Figure 3- please don't obstruct labelling of axes with their title, also was any sort of linear regression applied to this? What is the CC of the relationship – these numbers are useful in assessing the relationship that is not always obvious visually.

Figure 4 – I am really confused by this figure – there must be a better way of presenting these data. Line 400 what is a negative structure??

Line 404 – Fewer!

Line 429 – sentence starting 'generating' makes no sense at end.

Figure 5 – no label of density – again simulated annealing omit maps would be best

Discussion:

Start of the discussion would be better in the Introduction

Line 568 – here and throughout manuscript numerous references to 'as discussed earlier' better to give section.

Line 573 Confounding?? Confusing?

Line 559 – remove 'rather'

Reviewer #3 (Remarks to the Author):

The manuscript by Prof. Hubbard and colleagues presents fragment screening of two proteins, PDHK2 and Hsp90, using the Xchem facility from Diamond Light source to rapidly go through several tens of crude reaction mixtures (CRM) with an automatic procedure. 498

crystals/datasets (without those used for establishing the protocols) were analyzed to draw the conclusions (and not "more than a 1000" as stated in the abstract). These structural results of the two test crystals, are compared to the results obtained after user intervention also called "manual" in the paper but also against the affinity measurements by SPR.

The use of CRMs has the main advantage increasing the throughput of fragment screening by removing a painful step of purification. This method is equivalent to the fragment-based cocktail crystallography which was first introduced in the late 90s and several companies worldwide based their business plan on this approach. The protein-ligand affinity is investigated in the light of dissociation constant (K_{off}) as previously demonstrated being the main parameter to improve ligand affinity. The determination of association constant (K_{on}) would have been an asset to strengthen the argumentation. One should also keep in mind that the crystal constraints would make the measurement in solution not valid. The PDHK2 and Hsp90 are ATPase enzyme both members of the GHKL family having in common their main domain folding. The representativeness is compromised by this choice and especially in the view of exploring the structure/activity relationship with an high throughput technology but this is mentioned in the conclusion. The central portion of the main results table (table S1) shows the color coded results as defined in the text from line 353 to 358 and tell if the protein is native or if the starting material or the product can be fitted in the electron density. None of these density is available to make judgement and the PDB ids are not mentioned either. A map correlation calculation between the observed density and the calculated density of both SM and PR would have been beneficial especially for map at low resolution .

Along the manuscript, some results are presented without demonstration or evidence. In sentence line 381-383, the plot from figure 5 is not explicit enough to give the conclusion drawn and a minimum of statistics like rmsd of false positive and false negative would have been more meaningful. A less important one but still to be explained is the number of liters of solvent and the amount of reagent which have been saved (line 516-518). When numbers are presented often it is difficult to understand where they come from. Line 289, "498 crystals" comes from 83 CRMs times 3 for the triplicate experiment time 2 proteins PDHK2 and Hsp90. In the same vein, the main results table is presented in figure 3 showing only CRMs with yield > 10%. If we look only at PDHK2 (top graph) CRM1 to CRM10 there are five dots (four PR and one SM) and the table S1 shows seven CRM with a yield > 10% (CRM 1, 4-8 and 10). From line 132 to 136, 328 different compounds were used to start with. The crystallographic ligand screening shows 163 in the first attempt, plus 16 in the second and 6 more in the last attempt. The sum makes 185 different compounds these subtract to 328 means 143 compound for which no structure was available and not 133 as stated in line 136.

Referees report on: Rapid optimisation of fragments and hits to lead compounds from screening of crude reaction mixtures

The manuscript by Baker and colleagues describes important work to improve the ‘hit to lead’ process in the development of therapeutic or probe compounds. The authors use a ‘crude reaction mix’ to select compounds with higher affinity than the initial lead using the binding of the target protein to pull the compound out. The results are assessed using X-ray crystallography and surface plasmon resonance and are amenable to a high-throughput approach. While I believe this work presents an important and new method in compound development and merits publication in this journal; I found the manuscript to be very poorly written, confusing and difficult to read. I have made detailed comments below but for example the use of the acronyms PR and SM throughout I found very frustrating – I had to constantly refer back to what they were – why not just write product and starting material? These are just 2 examples. The tables and figures are poorly made and difficult to understand and need to be re-thought – there is a lot of information here and it could be presented in a much more intuitive fashion. A final small concern is that while the resolution obtained for the product bound complexes should be sufficiently high for unambiguous ligand determination the density in the reports looks often quite difficult to assign. I am not implying a lack of due diligence but I feel that this is an important aspect for the work presented here – identification of a ligand from a large mix – I feel the authors should describe at length how well they can ascribe the density to a specific component and how mistakes are avoided.

I have a number of general comments below that need answering and a large number of edits, that relate mainly to the use of language, that are found in ‘Specific comments’.

Specific Comments:

Title: The title is a bit confusing – maybe remove ‘and’

Abstract: line 29 insert ‘the’ before ‘generation’

Line 31 replace enough with ‘sufficient’

Line 34 and 35 – the reaction mixtures are not assessed but their binding to the target proteins, XChem has a remarkable reputation but I would say that high-throughput is sufficient rather than ultra-high and I would myself describe it as a data collection facility or a structure determination facility.

Introduction:

Line 52 is it OK to site a blog? If so I would prefer that the specific entry is cited – I could not find what is referenced here

Line 55 – large literature? A wide body of research?

Line 68 – prefer hundreds to thousands

Line 100 remove ‘this’

Line 102 – remove ‘ultra’

Line 106 and 107 replace hereafter with referred to as

It would be useful to expand on what exactly CRMs are here – they need to be properly introduced – maybe this is already well established but the legend for Table S1 seems to explain it so why not the intro? Also, The next section ‘Historical data’ is this still the introduction? Would be better in the results or methods. The introduction left me very unclear about what the point of the experiments were – the 2 paragraphs of the discussion expressed this very well – maybe move them here?

Results

Figure 1 Would be very good to have simulated annealing omit maps

Line 226 unknown reasons rather than the inverse.

Table S1 – there has obviously been a lot of effort put into this table but I found it so hard to interpret – for a start maybe better to separate the 2 proteins? Would be better to have these data in the main paper!

Line 259 – data were!

Line 271 – Do you mean by lower quality that because the active site contains a bound ligand you get a worse result to put all electron density on an absolute scale? I am not really sure why PanDDA is used here – it is excellent in being able to identify

Line 289 – single data set collection – do you mean a single experimental session?

Line 305 – would prefer column numbers or something more descriptive

Line 341 – products could be placed into manually assessed electron density maps?

Figure 3- please don't obstruct labelling of axes with their title, also was any sort of linear regression applied to this? What is the CC of the relationship – these numbers are useful in assessing the relationship that is not always obvious visually.

Figure 4 – I am really confused by this figure – there must be a better way of presenting these data. Line 400 what is a negative structure??

Line 404 – Fewer!

Line 429 – sentence starting 'generating' makes no sense at end.

Figure 5 – no label of density – again simulated annealing omit maps would be best

Discussion:

Start of the discussion would be better in the Introduction

Line 568 – here and throughout manuscript numerous references to 'as discussed earlier' better to give section.

Line 573 Confounding?? Confusing?

Line 559 – remove 'rather'

Rapid optimisation of fragments and hits to lead compounds from screening of crude reaction mixtures

7th July 2020

Response to reviewer's comments.

The comments from the reviewers were, for the most part, extremely helpful in identifying where the explanations provided in the text could be improved.

I restate here the comments of the reviewer in italics; with our response in bold

Reviewer #1 (Remarks to the Author):

While fragment-based drug discovery is a widely used alternative to high-throughput methods for initial lead identification, the development of fragments into high potent leads remains challenging. Here, the authors describe a method to efficiently screen for additional compounds with improved affinity by using crude reaction mixtures. This enables the detection of a vast number of additional binders in a material and time efficient manner.

I recommend publication after some minor corrections.

On page 7, line 136: Instead of 133 compounds it should be 143 compounds.

Agreed

*On page 9, Figure 1 and page 25, Figure 5: The electron density should be depicted as difference electron density (omit-map) at an appropriate level. We had shown the 2Fo-Fc density as this is the final map generated by the PanDDA process from which a fit metric is generated (now included in Table S1). This is now explained in more detail in the Methods section. Equivalent omit maps have been added as Figure S6. At least the PDB codes should be given here **Inserted** and the refinement statistics should be referred to the suppl. Material **Reference to Table S2 now included in Methods** Also, I assume that in Table S2 the complexes PDHK2 +3 and PDHK2 +4 refer to PDHK2 +23 and PDHK2 +24. **Corrected***

*On page 9, Figure 1: I found it interesting that compound SM2 showed improved affinity, despite the fact that there seem to be no additional interactions of SM2 to the protein when compared to the SM1 complex. The SM2 additional part (when compared to SM1) seems to stick into the solvent. Maybe the authors can comment on that. **Only selected protein residues are shown in Figure 1, but we chose a few to show additional interactions from SM2 - to Asp 54 in Hsp90 and Arg 250 in PDHK2 with some others through solvent to the protein. Those interested in more detail can look at the deposited structures.***

Page 27, last paragraph: Just out of curiosity, how does the hit rate improve if taken the results from only two experiments instead of three? Maybe 2 is the way to go?

Looking at the duplicate improves the results somewhat, as you would expect. The data for considering duplicates is now included in Table 4

Page 36: Author contributions: Include Frank von Delfts contribution to the paper.

Omission corrected

Reviewer #2 (Remarks to the Author):

*The manuscript by Baker and colleagues describes important work to improve the ‘hit to lead’ process in the development of therapeutic or probe compounds. The authors use a ‘crude reaction mix’ to select compounds with higher affinity than the initial lead using the binding of the target protein to pull the compound out. The results are assessed using X-ray crystallography and surface plasmon resonance and are amenable to a high-throughput approach. While I believe this work presents an important and new method in compound development and merits publication in this journal; I found the manuscript to be very poorly written, confusing and difficult to read. I have made detailed comments below but for example the use of the acronyms PR and SM throughout I found very frustrating – I had to constantly refer back to what they were – why not just write product and starting material? **We had hoped that the way we introduced the definition of PR and SM in the text would have defined the terms in the reader’s mind. A sentence has been added to reinforce this. PR means that the electron density from diffraction data collected for a protein crystal soaked with a crude reaction mixture in the XChem facility shows the expected product bound in the protein and, SM is the equivalent for starting material bound. Saying that every time would make the manuscript very long and tedious. Saying just “product” would be confusing as it does not differentiate from the products in the reaction mixtures themselves, or the structures where pure products were soaked into crystals. There would similarly be confusion if the phrase “starting material” was used. These are just 2 examples. The tables and figures are poorly made and difficult to understand and need to be re-thought – there is a lot of information here and it could be presented in a much more intuitive fashion. I appreciate that the ideas and that the amount of data we are analysing and distilling is quite different from any other study. We took a lot of time thinking about how to present the data in as simple a way as possible. From the comments of the reviewer, we can see that further explanation is required. A more detailed explanation of Table S1 is provided in the legend to that figure. A more detailed explanation of the CRMs is provided in the introduction and the beginning of the results.***

The plots have been redrawn with addition of some statistics in the accompanying legends.

*A final small concern is that while the resolution obtained for the product bound complexes should be sufficiently high for unambiguous ligand determination the density in the reports looks often quite difficult to assign. **We have added more explanation on how the decision was made to ascribe density as SM or PR and the challenges there are in using a metric to differentiate. I am not implying a lack of due diligence but I feel that this is an important aspect for the work presented here – identification of a ligand from a large mix This last comment (perhaps mis-written?) suggests the reviewer has misunderstood the experiments. The crude reaction mixtures are not a mixture of compounds – they are a mixture of a single starting material, reaction reagents and some amount of a single product where the identity of the starting material and putative product are known. – I feel the authors should describe at length how well they can ascribe the density to a specific component and how mistakes are avoided. Comment dealt with earlier in this paragraph.***

I have a number of general comments below that need answering and a large number of edits, that relate mainly to the use of language, that are found in 'Specific comments'.

Specific Comments:

Title: The title is a bit confusing – maybe remove 'and' The title is carefully chosen. The first sets of CRMs are elaborating a fragment, the second set of CRMs are elaborating a hit compound derived from a fragment – so saying “fragments and hits to leads” is what it is.

Abstract: line 29 insert 'the' before 'generation' done

Line 31 replace enough with 'sufficient' done

Line 34 and 35 – the reaction mixtures are not assessed but their binding to the target proteins. XChem has a remarkable reputation but I would say that high-throughput is sufficient rather than ultra-high and I would myself describe it as a data collection facility or a structure determination facility. XChem is not just data collection or structure determination it is both and more. It includes crystal soaking, mounting and refinement. So I think assess is OK. The use of the word “ultra” is a matter of opinion – I have removed it.

Introduction:

Line 52 is it OK to site a blog? If so I would prefer that the specific entry is cited – I could not find what is referenced here. Now corrected

Line 55 – large literature? A wide body of research? I think literature. There can be a lot of research without literature

Line 68 – prefer hundreds to thousands. If you are screening a fragment library containing thousands of compounds, then you need thousands of diffraction data sets.

Line 100 remove 'this' Done

Line 102 – remove 'ultra' Done

Line 106 and 107 replace hereafter with referred to as. I have inserted “referred to as”. It must be “hereafter referred to as” as before this sentence, Hsp90 and PDHK2 meant something else.

It would be useful to expand on what exactly CRMs are here – they need to be properly introduced – maybe this is already well established but the legend for Table S1 seems to explain it so why not the intro? Also, The next section 'Historical data' is this still the introduction? Would be better in the results or methods. The introduction left me very unclear about what the point of the experiments were – the 2 paragraphs of the discussion expressed this very well – maybe move them here? Good points. Some rearrangement and addition of text expands on the concept of CRMs where they are first mentioned in the introduction.

Results:

Figure 1 Would be very good to have simulated annealing omit maps We have now provided omit maps in the SI. Simulated annealing omit maps are not available in the CCP4 suite which is the software to which Vernalis has a license.

Line 226 unknown reasons rather than the inverse Done

Table S1 – there has obviously been a lot of effort put into this table but I found it so hard to interpret – for a start maybe better to separate the 2 proteins? Would be better to have these data in the main paper! The reviewer is correct to recognize

that a lot of thought (as well as effort) went into this table. The legend explains succinctly what each column contains and explains the colour coding. I can see that if the reviewer cannot hold in mind what PR and SM mean (comment above), then it will be a challenge to follow. Hopefully, emphasizing the importance of these terms will help. Separating the two proteins would mean a lot of duplication and I do not see how that would help the reviewer understand the table. I have added further explanation to the legend to the Table in the SI. We considered placing the table in the main text, but we realized it is not needed there as what is of most interest are the key results which we summarize in the main text. The table S1 provides (albeit condensed) detail of the results of all the experiments so it needs to be available but can be in supplementary information to support the summaries.

Line 259 – data were! **corrected**

Line 271 – Do you mean by lower quality that because the active site contains a bound ligand you get a worse result to put all electron density on an absolute scale?

The wording has been improved – the bound ligands in some of the datasets results in the binding site having some density noise in the averaged map. I am not really sure why PanDDA is used here – it is excellent in being able to iden, Although the crystallographer on the project heroically solved and refined all the structures manually, PanDDA is an essential component of the XChem explorer (XCE) system for handling all the data and routinely solving 1000s of crystal structures a week. Hence, we were keen to benchmark it against manual fitting.

Line 289 – single data set collection – do you mean a single experimental session?

Modified to “single dataset collection session” – the experiments were more than data collection – making crystals, picking crystals, soaking crystals, analysing data were all done at different times. The diffraction data was all collected in one session.

Line 305 – would prefer column numbers or something more descriptive **Added column numbers in the text, in Table S1 and in the legend to the table.**

Line 341 – products could be placed into manually assessed electron density maps? **the text has been modified to explain this more clearly.**

Figure 3- please don't obstruct labelling of axes with their title, also was any sort of linear regression applied to this? What is the CC of the relationship – these numbers are useful in assessing the relationship that is not always obvious visually. **The axes of the plots have been redrawn. A correlation coefficient is not appropriate – these are the range of 1/k_{off} values for one of two states. I have inserted mean and standard deviation to show there is no difference and used the Kolmogorov-Smirnoff test to calculate the probability that the values (1/k_{off}, nRot or HAC) for SM and PR for each protein were not from the same distribution.**

Figure 4 – I am really confused by this figure – there must be a better way of presenting these data. Line 400 what is a negative structure?? **This is explained in the text, perhaps too concisely. The wording has been changed to hopefully explain this more clearly.**

Line 404 – Fewer! **Corrected**

Line 429 – sentence starting ‘generating’ makes no sense at end. I think the reviewer misread. The sentence is: “22 of the 48 CRMs generating a product with slower k_{off} than SM1 gave SM.” **This scans correctly.**

Figure 5 – no label of density – again simulated annealing omit maps would be best **We have now provided omit maps in the SI. Simulated annealing omit maps are**

not available in the CCP4 suite which is the software to which Vernalis has a license.

Discussion:

Start of the discussion would be better in the Introduction; a good suggestion – done.
Line 568 – here and throughout manuscript numerous references to ‘as discussed earlier’ better to give section. There were two examples of ‘discussed earlier’ and two of ‘discussed above’. These have been modified, referring to sections where possible.

Line 573 Confounding?? Confusing? I think confounding is correct – it covers confusing. One definition is “cause surprise or confusion in (someone), especially by not according with their expectations” which is what I mean

Line 559 – remove ‘rather’ Removed. We were showing some modesty. It was exhausting as well as exhaustive working through not only the results, but equally trying to understand the confounding results.

Reviewer #3 (Remarks to the Author):

*The manuscript by Prof. Hubbard and colleagues presents fragment screening **This is not fragment screening. It is evolution of fragment and hit compounds** of two proteins, PDHK2 and Hsp90, using the Xchem facility from Diamond Light source to rapidly go through several tens of crude reaction mixtures (CRM) with an automatic procedure. 498 crystals/datasets (without those used for establishing the protocols) were analyzed to draw the conclusions (and not "more than a 1000" as stated in the abstract). **This phrase has been removed in a shortened abstract to meet the 150 word limit.** These structural results of the two test crystals, are compared to the results obtained after user intervention also called "manual" in the paper but also against the affinity measurements by SPR.*

*The use of CRMs has the main advantage increasing the throughput of fragment screening by removing a painful step of purification. This method is equivalent to the fragment-based cocktail crystallography which was first introduced in the late 90s and several companies worldwide based their business plan on this approach. **If historical comparisons are to be made, then this is equivalent to the very first experiments where compounds were soaked into crystals – as all crystal soaking experiments are selecting a compound that binds from a solution which contains other components from the crystallization liquor (and water). Reference to fragment cocktail soaking would be misleading as (at least in the early days) much larger mixtures of compounds were used, none of which were known to bind. Here we have a mixture of 2 compounds – a starting material which is known to bind and a putative product which may bind with higher affinity.***
*The protein-ligand affinity is investigated in the light of dissociation constant (K_{off}) as previously demonstrated being the main parameter to improve ligand affinity. The determination of association constant (K_{on}) would have been an asset to strengthen the argumentation. **Agreed – but the association constant is dependent on concentration. The key feature of the approach is that the off-rate is concentration independent. To measure k_{on} would require purification of the product and solutions made of known concentration. This point is emphasized in the revised introduction. One should also keep in mind that the crystal constraints***

would make the measurement in solution not valid. **The different solvent conditions in the crystal will affect the affinity, as will any constraints on conformational change (though some can be seen in crystals on compound binding). Other than that, the crystal environment itself may affect the kinetics of binding. Some additional sentences are added to the introduction to make this point. The PDHK2 and Hsp90 are ATPase enzyme both members of the GHKL family having in common their main domain folding. The representativeness is compromised by this choice and especially in the view of exploring the structure/activity relationship with an high throughput technology but this is mentioned in the conclusion. Agreed – these two systems were chosen as the chemistry and systems were already established for our previous study – but has the added advantage of demonstrating that the CRMs can identify selective compounds for these quite similar binding sites. The central portion of the main results table (table S1) shows the color coded results as defined in the text from line 353 to 358 and tell if the protein is native or if the starting material or the product can be fitted in the electron density. None of these density is available to make judgement and the PDB ids are not mentioned either. A map correlation calculation between the observed density and the calculated density of both SM and PR would have been beneficial especially for map at low resolution/. We have added some metrics on fit to density and included a discussion in the supplementary information on calculations which demonstrate it is not possible to use the metrics to make the decision between SM and PR.**

Along the manuscript, some results are presented without demonstration or evidence. In sentence line 381-383, the plot from figure 5 (**Figure S5**) is not explicit enough to give the conclusion drawn and a minimum of statistics like rmsd of false positive and false negative would have been more meaningful. **I thank the reviewer for raising this. I have inserted mean and standard deviation to in the legend to show there is little or no difference and used the Kolmogorov-Smirnoff test to calculate the probability that the values ($1/k_{\text{off}}$, nRot or HAC) for SM and PR for each protein were or were not from the same distribution. This suggested that compounds with an increased number of rotatable bonds with slower k_{off} were slightly more likely to give SM from the CRM – the difference is slight and soaking with pure compound does give product bound. A few sentences have been added in the results and discussion session and the statistics and notched box plots included in SI. A less important one but still to be explained is the number of liters of solvent and the amount of reagent which have been saved (line 516-518). When numbers are presented often it is difficult to understand where they come from. A substantial section has been added to the SI to explain how these numbers were estimated. Line 289, "498 crystals" comes from 83 CRMs times 3 for the triplicate experiment time 2 proteins PDHK2 and Hsp90. In the same vein, the main results table is presented in figure 3 showing only CRMs with yield > 10%. If we look only at PDHK2 (top graph) CRM1 to CRM10 there are five dots (four PR and one SM) and the table S1 shows seven CRM with a yield > 10% (CRM 1, 4-8 and 10). The reviewer has counted the dots correctly - however for CRMs 4 and 7, k_{off} was not determined (ND) – so there is no k_{off} value and the plot is correct. The legend has been modified to make this clear. From line 132 to 136, 328 different compounds were used to start with. The crystallographic ligand screening shows 163 in the first attempt, plus 16 in the second and 6 more in the last attempt. The sum makes 185 different compounds these subtract to 328 means 143 compound for**

which no structure was available and not 133 as stated in line 136. **Thank you for checking the arithmetic – number corrected.**

REVIEWERS' COMMENTS:

Reviewer #1 (Remarks to the Author):

The authors have appropriately addressed the comments previously made by Reviewer 1. As there were only minor corrections required, the manuscript may now be published.

The only remaining correction I have is in Table 4, PDHK2. Here the number from singleton PR was changed in the new version to "34", while it was "40" in the initial manuscript. In the manuscript text it is still "40". Please enter the correct number.

Reviewer #2 (Remarks to the Author):

The authors have addressed all points I raised satisfactorily.

Reviewer #3 (Remarks to the Author):

The authors corrected and modified the manuscript accordingly. The major concerns have been addressed.

Rapid optimisation of fragments and hits to lead compounds from screening of crude reaction mixtures

Final comment from reviewer:

The only remaining correction I have is in Table 4, PDHK2. Here the number from singleton PR was changed in the new version to "34", while it was "40" in the initial manuscript. In the manuscript text it is still "40". Please enter the correct number

The correct number is 34 which is now consistent in the table and the main text.